# Dominated Actions in Imperfect-Information Games

## Abstract

Dominance is a fundamental concept in game theory. In strategic-form games dominated strategies can be identified in polynomial time. As a consequence, iterative removal of dominated strategies can be performed efficiently as a preprocessing step for reducing the size of a game before computing a Nash equilibrium. For imperfect-information games in extensive form, we could convert the game to strategic form and then iteratively remove dominated strategies in the same way; however, this conversion may cause an exponential blowup in game size. In this paper we define and study the concept of dominated actions in imperfect-information games. Our main result is a polynomial-time algorithm for determining whether an action is dominated (strictly or weakly) by any mixed strategy in $n$-player games, which can be extended to an algorithm for iteratively removing dominated actions. This allows us to efficiently reduce the size of the game tree as a preprocessing step for Nash equilibrium computation. We explore the role of dominated actions empirically in the "All In or Fold" No-Limit Texas Hold'em poker variant.

## 1 Introduction

In a strategic-form game, (mixed) strategy $\sigma_i$ for player $i$ is *strictly dominated* if there exists another (mixed) strategy $\sigma_i'$ such that $\sigma_i'$ performs strictly better than $\sigma_i$ regardless of the strategy used by the opponent(s): formally, if $u_i(\sigma_i', s_{-i}) > u_i(\sigma_i', s_{-i})$ for all pure strategy profiles (vector of pure strategies) $s_{-i} \in S_{-i}$ for the opposing agents. (Note that the requirement that this holds for all opposing pure strategy profiles is enough to ensure that it also holds for all mixed strategy profiles of the opponents as well). Clearly it would be irrational for an agent to play a strictly dominated strategy, as another strategy will do strictly better regardless of the beliefs of what the opponents would play. Strategy $\sigma_i'$ *weakly dominates* $\sigma$ if the inequality holds weakly (though strictly for at least one pure strategy profile): formally, $u_i(\sigma', s_{-i}) \geq u_i(\sigma_i', s_{-i})$ for all $s_{-i} \in S_{-i}$ where the inequality is strict for at least one $s_{-i}$. (The condition that at least one inequality is strict is simply to rule out saying that a strategy is dominated by an identical strategy). Similar to strict domination it seems clearly irrational for an agent to play a strategy that is weakly dominated, as another strategy will perform at least as well (and sometimes strictly better) regardless of the strategies employed by the opponent(s).

It seems natural to simplify a game by eliminating strategies that are dominated from the game to reduce its size and focus analysis on a smaller game. It can easily be shown that applying an iterative process of removing one dominated strategy for one player, then removing one for another (or possibly the same) player in the reduced game, etc., will ultimately result in a smaller game that contains a Nash equilibrium from the original game. For this reason, this procedure of *iterated removal of dominated strategies* is often performed as a preprocessing step to reduce the size of a game before computing a Nash equilibrium (or other desired solution concept). As it turns out, all processes of iterated removal of strictly dominated strategies produce the same reduced game regardless of the order of elimination (while this is not necessarily the case for iterated removal of weakly dominated strategies) [5, 8]. While iterated removal of weakly dominated strategies can

sometimes reduce the number of equilibria, it can never create new equilibria, and therefore even this procedure is very useful as a preprocessing step for Nash equilibrium computation.

There is a linear-time algorithm for determining whether a (mixed) strategy $\sigma_i$ is strictly dominated by any pure strategy for player $i$ [12]. This algorithm simply iterates over each pure strategy $s_i$ for player $i$ and tests whether it performs strictly better than $\sigma_i$ against each opposing pure strategy profile $s_{-i}$. This procedure has complexity $O(|A|)$, where $A = \times_i A_i$ is the set of joint action (i.e., pure strategy) profiles, and so takes time linear in the size of the game. The procedure can also be easily adapted to produce an algorithm for determining whether a mixed strategy profile is weakly dominated by any pure strategy in linear time. Note that it is possible for a strategy to be dominated by a mixed strategy and not be dominated by any pure strategy [12]. In order to determine whether a (mixed) strategy is strictly dominated by a mixed strategy, while the above procedure does not work, it turns out that there exists a linear programming formulation that runs in polynomial time, and there also exists a linear programming formulation that determines whether a (mixed) strategy is weakly dominated by a mixed strategy [1]. Therefore, regardless of whether the strategies being tested as dominated or dominating are mixed or pure, it can be checked in polynomial time.[1]

## 2 Extensive-form games

While the strategic form can be used to model simultaneous actions, another representation, called the *extensive form*, is generally preferred when modeling settings that have sequential moves. The extensive form can also model simultaneous actions, as well as chance events and imperfect information (i.e., situations where some information is available to only some of the agents and not to others). Extensive-form games consist primarily of a game tree; each non-terminal node has an associated player (possibly *chance*) that makes the decision at that node, and each terminal node has associated utilities for the players. Additionally, game states are partitioned into *information sets*, where the player whose turn it is to move cannot distinguish among the states in the same information set. Therefore, in any given information set, a player must choose actions with the same distribution at each state contained in the information set. If no player forgets information that they previously knew, we say that the game has *perfect recall*. A (behavioral) *strategy* for player $i$, $\sigma_i \in \Sigma_i$, is a function that assigns a probability distribution over all actions at each information set belonging to $i$.

In theory, every extensive-form game can be converted to an equivalent strategic-form game; however, there is an exponential blowup in the size of the game representation, and therefore such a conversion is undesirable. Instead, new algorithms have been developed that operate on the extensive form representation directly. It turns out that the complexity of computing equilibria in extensive-form games is similar to that of strategic-form games; a Nash equilibrium can be computed in polynomial time in two-player zero-sum games (with perfect recall) [7], while the problem is hard for two-player general-sum and multiplayer games. One algorithm for computing an equilibrium in two-player zero-sum extensive-form games with perfect recall is based on solving a linear programming formulation [7]. This formulation works by modeling each sequence of actions for each player as a variable, and is often called the *sequence form LP* algorithm. Note that while the number of pure strategies is exponential in the size of the game tree, the number of action sequences is linear. The method uses several matrices defined as follows. For player 1, the matrix $\mathbf{E}$ is defined where each row corresponds to an information set (including an initial row for the "empty" information set), and each column corresponds to an action sequence (including an initial row for the "empty" action sequence). In the first row of $\mathbf{E}$ the first element is 1 and all other elements are 0; subsequent rows have -1 for the entries corresponding to the action sequence leading to the root of the information set, and 1 for all actions that can be taken at the information set (and 0 otherwise). Thus $\mathbf{E}$ has dimension $c_1 \times d_1$, where $c_i$ is the number of information sets for player $i$ and $d_i$ is the number of action sequences for player $i$. Matrix $\mathbf{F}$ is defined analogously for player 2. The vector $\mathbf{e}$ is defined to be a column vector of length $c_1$ with 1 in the first position and 0 in other entries, and vector $\mathbf{f}$ is defined with length $c_2$ analogously. The matrix $\mathbf{A}$ is defined with dimension $d_1 \times d_2$ where entry $A_{ij}$ gives the expected payoff for player 1 when player 1 plays action sequence $d_1$ and player 2 plays action sequence $d_2$

---

[1]However, most other computational questions related to dominance can be solved in polynomial time for strict but are NP-hard for weak dominance. For example, determining whether there exists some elimination path under which strategy $s_i$ is eliminated, given action subsets $A_i' \subset A_i$ determining whether there exists a maximally reduced game where each player has the actions $A_i'$, and given constants $k_i$ determining whether there exists a maximally reduced game where each player has exactly $k_i$ actions [12, 6].

89  (with the expectation being over possible moves of chance along the paths of play to leaf nodes). The
90  matrix $\mathbf{B}$ of player 2's expected payoffs is defined analogously. In zero-sum games $\mathbf{B} = -\mathbf{A}$.

91  Given these matrices we can solve one of two linear programming problems to compute a Nash
92  equilibrium in zero-sum extensive-form games [7]. In the first formulation the primal variables $\mathbf{x}$
93  correspond to player 1's mixed strategy while the dual variables correspond to player 2's strategy.
94  In the second formulation, which is the dual problem of the first formulation, the primal decision
95  variables $\mathbf{y}$ correspond to player 2's strategy while the dual variables correspond to player 1's strategy.

$$
\begin{aligned}
\max_{\mathbf{x},\mathbf{q}} \quad & -\mathbf{q}^T\mathbf{f} \\
\text{s.t.} \quad & \mathbf{x}^T(-\mathbf{A}) - \mathbf{q}^T\mathbf{F} \le \mathbf{0} \\
& \mathbf{x}^T\mathbf{E}^T = \mathbf{e}^T \\
& \mathbf{x} \ge \mathbf{0}
\end{aligned}
$$

$$
\begin{aligned}
\min_{\mathbf{y},\mathbf{p}} \quad & \mathbf{e}^T\mathbf{p} \\
\text{s.t.} \quad & -\mathbf{A}\mathbf{y} + \mathbf{E}^T\mathbf{p} \ge \mathbf{0} \\
& -\mathbf{F}\mathbf{y} = -\mathbf{f} \\
& \mathbf{y} \ge \mathbf{0}
\end{aligned}
$$

## 3  Dominated actions

97  In extensive-form games, we can consider analogous concepts of strict, weak, and iterated dominance
98  of strategies as for strategic-form games. However, unlike in the strategic-form setting, identification
99  of a dominated extensive-form strategy does not necessarily allow us to reduce the size of the game,
100  since it is possible that some of the actions played by the dominated strategy are also played by
101  non-dominated strategies. In order to obtain the computational advantage of game size reduction, we
102  must consider a stronger concept of *dominated actions*. We first present several plausible candidate
103  definitions for dominated actions which we demonstrate to be problematic. Our first candidate
104  definition is given as Flawed Definition 1. This definition states that action $a_i$ for player $i$ at
105  information set $I_i$ is strictly dominated by action $b_i$ at the same information set if every leaf node
106  succeeding $a_i$ produces a strictly smaller payoff for player $i$ than every leaf node succeeding action
107  $b_i$. An analogous definition for weak dominance is given in Flawed Definition 2.

108  **Flawed Definition 1.** *If for every leaf node $N^{a_i}$ that follows action $a_i$ for player $i$ at information*
109  *set $I_i$ and every leaf node $N^{b_i}$ that follows action $b_i$ for player $i$ at the same information set $I_i$,*
110  $u_i\left(N^{b_i}\right) > u_i\left(N^{a_i}\right)$, *then $b_i$ strictly dominates $a_i$.*

111  **Flawed Definition 2.** *If for every leaf node $N^{a_i}$ that follows action $a_i$ for player $i$ at information*
112  *set $I_i$ and every leaf node $N^{b_i}$ that follows action $b_i$ for player $i$ at the same information set $I_i$,*
113  $u_i\left(N^{b_i}\right) \ge u_i\left(N^{a_i}\right)$ *where inequality is strict for at least one node, then $b_i$ weakly dominates $a_i$.*

114  The problem with these definitions are that they are too strong; it is still possible for player 1 to strictly
115  prefer to take action $b_i$ to $a_i$ regardless of the strategy used by player 2 even if Flawed Definition 1
116  holds. This is illustrated in the following game depicted in Figure 1. In this game, chance makes an
117  initial move taking each of two actions with probability $\frac{1}{2}$. Then player 1 (red) selects one of two
118  actions at a single information set. Player 2 (blue) then takes one of two actions after observing both
119  the moves of chance and player 1. It is clear that action 2 for player 2 at their top information set and
120  action 1 for player 2 at their bottom information set are both strictly dominated according to Flawed
121  Definition 1. The smaller game obtained after removing these actions is depicted in Figure 2. In the
122  smaller game, the expected utility of playing action 1 is $0.5(-100) + 0.5(100) = 0$ and the expected
123  utility of playing action 2 is $0.5(-50) + 0.5(-50) = -50$. Since player 2 does not take any actions,
124  player 1 always achieves a strictly higher expected utility by playing action 1. However, action 2 is
125  not strictly dominated according to Flawed Definition 1 because in one leaf node succeeding action 1
126  player 1 obtains payoff -100 which is lower than the payoff of -50 at leaf nodes following action 2.

127  Flawed Definitions 1 and 2 provide sufficient conditions for an action to be dominated, but we have
128  demonstrated that it is clearly possible for actions that do not satisfy these definitions to also be
129  dominated. Thus these conditions are too strong, and we will refer to strategies that satisfy them
130  as being *strongly dominated* (strictly or weakly, respectively). The concept of strong dominance
131  is not without merit, as it can be verified very efficiently by simply iterating over the leaf nodes

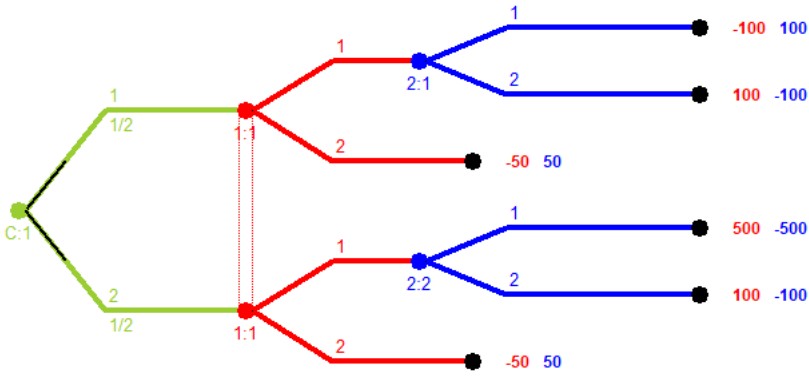

Figure 1: Example two-player imperfect-information extensive-form game.

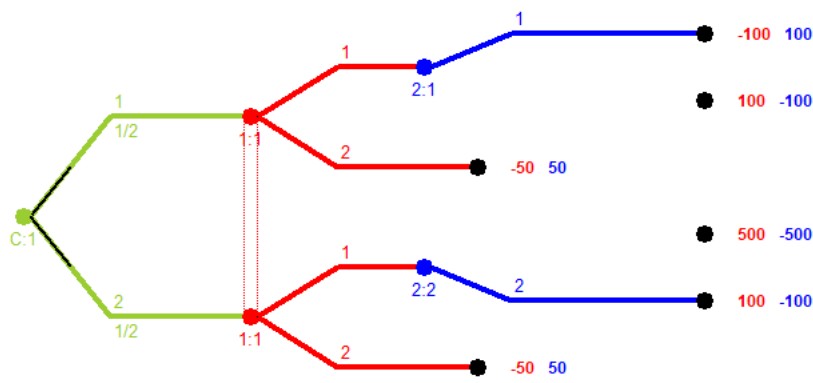

Figure 2: Result of removing two dominated actions from game in Figure 1.

succeeding each action. Thus it may be useful to first remove actions that are strongly dominated as a preprocessing step before performing potentially more costly computations to remove other actions.

We next consider candidate definition given by Flawed Definition 3, where $u_i$ denotes the expected utility accounting for randomized moves of chance as well as potential randomization in the players' strategies. This definition states that action $a_i$ for player $i$ at information set $I_i$ is strictly dominated (potentially by a probability distribution over other actions at the same information set), if there exists a strategy $\sigma_i^{-a_i}$ that never plays $a_i$ at $I_i$ that always has strictly higher expected utility than every strategy that plays $a_i$ at $I_i$. Again this definition clearly provides a sufficient condition for $a_i$ to be dominated; however, the issue with this definition is that the strategies may potentially take actions early in the game tree that prevent the game from ever reaching information set $I_i$. Consider the simple example game in Figure 3. Suppose we want to apply Flawed Definition 3 to determine whether action 2 is strictly dominated for player 2. Then $\sigma_i^{-a_i}$ is the strategy for player 2 that plays action 1 with probability 1, and $\sigma_i^{a_i}$ must be the strategy for player 2 that plays action 2 with probability 1. Now suppose that player 1 plays action 2 with probability 1, which we denote as $\sigma_{-i}$. Then clearly $u_i\left(\sigma_i^{-a_i}, \sigma_{-i}\right) = u_i\left(\sigma_i^{a_i}, \sigma_{-i}\right) = 0$, since both strategy profiles will result in reaching the bottom leaf node yielding payoff 0.

**Flawed Definition 3.** *Action $a_i$ for player $i$ is* strictly dominated *at information set $I_i$ if there exists a mixed strategy $\sigma_i^{-a_i}$ that plays action $a_i$ at $I_i$ with probability 0 such that for every mixed strategy $\sigma_i^{a_i}$ for player $i$ that plays action $a_i$ with probability 1 at $I_i$, $u_i\left(\sigma_i^{-a_i}, \sigma_{-i}\right) > u_i\left(\sigma_i^{a_i}, \sigma_{-i}\right)$ for all opposing strategies $\sigma_{-i} \in \Sigma_{-i}$.*

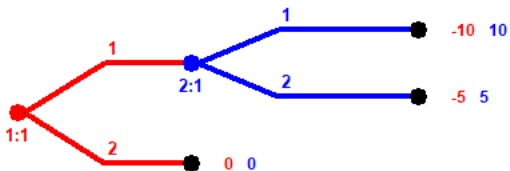

Figure 3: Extensive-form game demonstrating problem with Flawed Definition 3.

The problem with Flawed Definition 3 is that it allows the players to deviate from the path of play leading to the relevant information set $I_i$. We address this limitation in our new definitions. *Strictly-dominated actions* are defined in Definition 1 and *weakly-dominated actions* are defined in Definition 2. In these definitions $\Sigma_{-i}^{I_i}$ denotes the set of mixed strategy profiles for the opponents that always take actions leading to information set $I_i$ when possible. Note that these definitions apply to games with any number of players. They also consider actions that are dominated by any mixed strategy (not necessarily just a pure action at $I_i$). It is easy to verify that these definitions address the issues that arose in the examples from Figure 1 and Figure 3. We can apply these definitions repeatedly in succession to perform iterative removal of dominated actions, as for the strategic form.

**Definition 1.** *Action $a_i$ for player $i$ is* strictly dominated *at information set $I_i$ if there exists a mixed strategy $\sigma_i^{-a_i}$ that always plays to get to $I_i$ and plays action $a_i$ at $I_i$ with probability 0 such that for every mixed strategy $\sigma_i^{a_i}$ for player $i$ that always plays to get to $I_i$ and plays action $a_i$ with probability 1 at $I_i$, $u_i\left(\sigma_i^{-a_i}, \sigma_{-i}\right) > u_i\left(\sigma_i^{a_i}, \sigma_{-i}\right)$ for all opposing strategies $\sigma_{-i} \in \Sigma_{-i}^{I_i}$.*

**Definition 2.** *Action $a_i$ for player $i$ is* weakly dominated *at information set $I_i$ if there exists a mixed strategy $\sigma_i^{-a_i}$ that always plays to get to $I_i$ and plays action $a_i$ at $I_i$ with probability 0 such that for every mixed strategy $\sigma_i^{a_i}$ for player $i$ that always plays to get to $I_i$ and plays action $a_i$ with probability 1 at $I_i$, $u_i\left(\sigma_i^{-a_i}, \sigma_{-i}\right) \geq u_i\left(\sigma_i^{a_i}, \sigma_{-i}\right)$ for all opposing strategies $\sigma_{-i} \in \Sigma_{-i}^{I_i}$, where the inequality is strict for at least one $\sigma_{-i}$.*

The example games considered were created using the open-source software package Gambit [11]. Gambit has tools that allow the user to remove "strictly dominated" or "strictly or weakly dominated" actions, and these procedures can be repeated to iteratively remove multiple actions sequentially; however, there is no documentation regarding the algorithms applied or definitions of dominance used. In the example game from Figure 1, Gambit correctly identifies that action 2 for player 2 at their top information set and action 1 for player 2 at their bottom information set are both strictly dominated, and removes these to construct the smaller game in Figure 2. However, as for strong domination, Gambit fails to recognize that action 2 is strictly dominated by action 1 for player 1 in the reduced game. This example demonstrates that Gambit's procedure does not remove all dominated actions (though it does not necessarily imply that Gambit is only removing strongly dominated actions).

# 4 Algorithm for identifying dominated actions

Suppose we want to determine whether an action $c$ for player 1 is dominated at information set $I$ in a two-player extensive-form game $G$. Using the sequence form representation, suppose that action $c$ is the final action in the sequence with index $i$ for player 1. Let $S_I$ denote the set of indices corresponding to action sequences leading to $I$. We would like to solve the following problem.

$$
\begin{aligned}
\max_{\mathbf{x}_2} \min_{\mathbf{x}_1, \mathbf{y}} \quad & \mathbf{x}_2^T \mathbf{A} \mathbf{y} - \mathbf{x}_1^T \mathbf{A} \mathbf{y} \\
\text{s.t.} \quad & \mathbf{x}_{1,i} = 1 \\
& \mathbf{x}_{2,i} = 0 \\
& \mathbf{x}_{1,k} = \mathbf{x}_{2,k} = 1 \text{ for all } k \in S_I \\
& \mathbf{y}_k = 1 \text{ for all } k \in S_I \\
& \mathbf{x}_1^T \mathbf{E}^T = \mathbf{e}^T \\
& \mathbf{x}_2^T \mathbf{E}^T = \mathbf{e}^T \\
& \mathbf{y}^T \mathbf{F}^T = \mathbf{f}^T \\
& \{\mathbf{x}_1, \mathbf{x}_2, \mathbf{y}\} \geq \mathbf{0}
\end{aligned}
\tag{1}
$$

Consider the following problem, where now player 2 controls two action sequences $\mathbf{y_1}, \mathbf{y_2}$:

$$
\begin{aligned}
\max_{\mathbf{x}_2} \min_{\mathbf{x}_1, \mathbf{y}_1, \mathbf{y}_2} \quad & \mathbf{x}_2^T \mathbf{A} \mathbf{y_2} - \mathbf{x}_1^T \mathbf{A} \mathbf{y_1} \\
\text{s.t.} \quad & \mathbf{x}_{1,i} = 1 \\
& \mathbf{x}_{2,i} = 0 \\
& \mathbf{x}_{1,k} = \mathbf{x}_{2,k} = 1 \text{ for all } k \in S_I \\
& \mathbf{y}_{1,k} = 1 \text{ for all } k \in S_I \\
& \mathbf{y}_{2,k} = 1 \text{ for all } k \in S_I \\
& \mathbf{x}_1^T \mathbf{E}^T = \mathbf{e}^T \\
& \mathbf{x}_2^T \mathbf{E}^T = \mathbf{e}^T \\
& \mathbf{y}_1^T \mathbf{F}^T = \mathbf{f}^T \\
& \mathbf{y}_2^T \mathbf{F}^T = \mathbf{f}^T \\
& \{\mathbf{x}_1, \mathbf{x}_2, \mathbf{y_1}, \mathbf{y_2}\} \geq \mathbf{0}
\end{aligned}
\tag{2}
$$

**Proposition 1.** *The optimal objective values in Problem 1 and Problem 2 are the same.*

*Proof.* Let $f_1$ be the optimal objective value in Problem 1 and $f_2$ be the optimal objective value in Problem 2. Suppose that the optimal variables in Problem 1 are $\mathbf{x}_1^1, \mathbf{x}_2^1, \mathbf{y}^1$. Now set $\mathbf{x}_1^2 = \mathbf{x}_1^1$, $\mathbf{x}_2^2 = \mathbf{x}_2^1$, $\mathbf{y}_1^2 = \mathbf{y}^1$, $\mathbf{y}_2^2 = \mathbf{y}^1$. Then $(\mathbf{x}_1^2, \mathbf{x}_2^2, \mathbf{y}_1^2, \mathbf{y}_2^2)$ gives a feasible solution to Problem 2 with objective value equal to $f_1$. So $f_2 \geq f_1$. Now suppose that the optimal variables in Problem 2 are $\mathbf{x}_1^2, \mathbf{x}_2^2, \mathbf{y}_1^2, \mathbf{y}_2^2$. Now set $\mathbf{x}_1^1 = \mathbf{x}_1^2$, $\mathbf{x}_2^1 = \mathbf{x}_2^2$, and set $\mathbf{y}^1$ equal to the strategy that follows $\mathbf{y}_2^2$ at states following action $c$ for player 1, and follows $\mathbf{y}_1^2$ otherwise. Then $(\mathbf{x}_1^1)^T \mathbf{A} \mathbf{y}^1 = (\mathbf{x}_1^1)^T \mathbf{A} \mathbf{y}_1^2$, since both players only take actions to get to information set $I$, $\mathbf{x}_1^1$ takes action $c$ at $I$, and the strategies $\mathbf{y}^1$ and $\mathbf{y}_1^2$ are identical after player 1 takes action $c$ at $I$. Similarly, $(\mathbf{x}_2^1)^T \mathbf{A} \mathbf{y}^1 = (\mathbf{x}_2^1)^T \mathbf{A} \mathbf{y}_2^2$, since both players only take actions to get to $I$, $\mathbf{x}_1^1$ does not take action $c$ at $I$, and the strategies $\mathbf{y}^1$ and $\mathbf{y}_2^2$ are identical after player 1 does not take action $c$ at $I$. So $f_1 \geq f_2$. So we conclude that $f_1 = f_2$. $\square$

Proposition 1 allows us to divide Problem 2 into the following two subproblems. If $f$ is the optimal objective value of Problem 2, $f_1$ is the optimal objective of Problem 3, and $f_2$ is the optimal objective of Problem 4, then we have $f = f_1 - f_2$.

$$
\begin{aligned}
\max_{\mathbf{x}_2} \min_{\mathbf{y}} \quad & \mathbf{x}_2^T \mathbf{A} \mathbf{y} \\
\text{s.t.} \quad & \mathbf{x}_{2,i} = 0 \\
& \mathbf{x}_{2,k} = 1 \text{ for all } k \in S_I \\
& \mathbf{y}_k = 1 \text{ for all } k \in S_I \\
& \mathbf{x}_2^T \mathbf{E}^T = \mathbf{e}^T \\
& \mathbf{y}^T \mathbf{F}^T = \mathbf{f}^T \\
& \{\mathbf{x}_2, \mathbf{y}\} \geq \mathbf{0}
\end{aligned}
\tag{3}
$$

$$
\begin{aligned}
\max_{\mathbf{x}_1} \max_{\mathbf{y}} \quad & \mathbf{x}_1^T \mathbf{A} \mathbf{y} \\
\text{s.t.} \quad & \mathbf{x}_{1,i} = 1 \\
& \mathbf{x}_{1,k} = 1 \text{ for all } k \in S_I \\
& \mathbf{x}_1^T \mathbf{E}^T = \mathbf{e}^T \\
& \mathbf{y}^T \mathbf{F}^T = \mathbf{f}^T \\
& \{\mathbf{x}_1, \mathbf{y}\} \geq \mathbf{0}
\end{aligned}
\tag{4}
$$

Let us first look at Problem 3 and consider the inner subproblem for a fixed $\mathbf{x}_2$.

$$
\begin{aligned}
\min_{\mathbf{y}} \quad & \mathbf{x}_2^T \mathbf{A} \mathbf{y} \\
\text{s.t.} \quad & \mathbf{y}_k = 1 \text{ for all } k \in S_I \\
& \mathbf{y}^T \mathbf{F}^T = \mathbf{f}^T \\
& \mathbf{y} \geq \mathbf{0}
\end{aligned}
$$

The Lagrangian is

$$
L(\mathbf{y}, \boldsymbol{\lambda}, \boldsymbol{\gamma}, \mathbf{r}) = \mathbf{x}_2^T \mathbf{A} \mathbf{y} - (\mathbf{f}^T - \mathbf{y}^T \mathbf{F}^T) \boldsymbol{\lambda} - \sum_{k \in S_I} \gamma_k (\mathbf{y}_k - 1) - \mathbf{r}^T \mathbf{y}
$$

$$\frac{\partial L}{\partial \mathbf{y}} = \mathbf{x}_2^T \mathbf{A} + \boldsymbol{\lambda}^T \mathbf{F} - \sum_{k \in S_I} \gamma_k \mathbf{e_k} - \mathbf{r}^T$$

201  The dual problem is

$$
\begin{aligned}
\max_{\boldsymbol{\lambda}, \boldsymbol{\gamma}} \quad & -\mathbf{f}^T \boldsymbol{\lambda} - \sum_{k \in S_I} \gamma_k \\
\text{s.t.} \quad & \mathbf{x}_2^T \mathbf{A} + \boldsymbol{\lambda}^T \mathbf{F} - \sum_{k \in S_I} \gamma_k \mathbf{e_k} \geq \mathbf{0}
\end{aligned}
$$

202  So Problem 3 is equivalent to:

$$
\begin{aligned}
\max_{\mathbf{x}_2, \boldsymbol{\lambda}, \boldsymbol{\gamma}} \quad & -\mathbf{f}^T \boldsymbol{\lambda} - \sum_{k \in S_I} \gamma_k \\
\text{s.t.} \quad & \mathbf{x}_2^T \mathbf{A} + \boldsymbol{\lambda}^T \mathbf{F} - \sum_{k \in S_I} \gamma_k \mathbf{e_k} \geq \mathbf{0} \\
& \mathbf{x}_{2,i} = 0 \\
& \mathbf{x}_{2,k} = 1 \text{ for all } k \in S_I \\
& \mathbf{x}_2^T \mathbf{E}^T = \mathbf{e}^T \\
& \mathbf{x}_2 \geq \mathbf{0}
\end{aligned} \tag{5}
$$

203  Next consider Problem 4. Note that both players are aligned in trying to maximize the objective. Let
204  us define a new problem $\overline{G}$ where player 1 selects the actions for both players. We denote player 1's
205  strategy in this modified game by $\overline{\mathbf{x}}_1$. We modify $\mathbf{E}$ and resize $\mathbf{e}$ accordingly and denote them by $\mathbf{E}'$
206  and $\mathbf{e}'$. The payoffs can now be represented as a vector $\mathbf{a}$. In the new representation let $i'$ denote the
207  index of the sequence with concluding action $c$, and let $S_I'$ denote the set of indices corresponding to
208  action sequences leading to $I$.

$$
\begin{aligned}
\max_{\overline{\mathbf{x}}_1} \quad & \overline{\mathbf{x}}_1^T \mathbf{a} \\
\text{s.t.} \quad & \overline{\mathbf{x}}_{1,i'} = 1 \\
& \overline{\mathbf{x}}_{1,k} = 1 \text{ for all } k \in S_I' \\
& \overline{\mathbf{x}}_1^T \mathbf{E}'^T = \mathbf{e}'^T \\
& \overline{\mathbf{x}}_1 \geq \mathbf{0}
\end{aligned} \tag{6}
$$

209  Let $u_2$ denote the optimal objective value for optimization problem 5, and $u_1$ denote the optimal
210  objective value for problem 6. If $u_2 > u_1$ then we conclude that action $c$ is strictly dominated. If
211  $u_2 < u_1$ then we conclude that action $c$ is not strictly or weakly dominated. If $u_2 = u_1$, let $u_3$ denote
212  the optimal objective value for problem 7, and let $u_4$ denote the optimal objective value for problem 8.
213  If $u_3 > u_4$ then we conclude that action $c$ is weakly dominated, and if $u_3 = u_4$ we conclude that
214  action $c$ is not strictly or weakly dominated (note that we cannot have $u_3 < u_4$).

$$
\begin{aligned}
\max_{\overline{\mathbf{x}}_2, \boldsymbol{\lambda}, \boldsymbol{\gamma}} \quad & \overline{\mathbf{x}}_2^T \mathbf{a} \\
\text{s.t.} \quad & -\mathbf{f}^T \boldsymbol{\lambda} - \sum_{k \in S_I} \gamma_k = u_2 \\
& \mathbf{x}_2^T \mathbf{A} + \boldsymbol{\lambda}^T \mathbf{F} - \sum_{k \in S_I} \gamma_k \mathbf{e_k} \geq \mathbf{0} \\
& \overline{\mathbf{x}}_{2,i'} = 0 \\
& \overline{\mathbf{x}}_{2,k} = 1 \text{ for all } k \in S_I' \\
& \overline{\mathbf{x}}_2^T \mathbf{E}'^T = \mathbf{e}'^T \\
& \overline{\mathbf{x}}_2 \geq \mathbf{0}
\end{aligned} \tag{7}
$$

The components of $\overline{\mathbf{x}}_2$ for player 1 in $\overline{G}$ correspond to $\mathbf{x}_2$ in $G$.

$$
\begin{aligned}
\min_{\overline{\mathbf{x}}_1} \quad & \overline{\mathbf{x}}_1^T \mathbf{a} \\
\text{s.t.} \quad & \overline{\mathbf{x}}_{1,i'} = 1 \\
& \overline{\mathbf{x}}_{1,k} = 1 \text{ for all } k \in S_I' \\
& \overline{\mathbf{x}}_1^T \mathbf{E}'^T = \mathbf{e}'^T \\
& \overline{\mathbf{x}}_1 \geq \mathbf{0}
\end{aligned} \tag{8}
$$

215  We can perform this procedure for every action at each information set of player 1, and analogously
216  for player 2. Since the number of actions is linear in the size of the game tree, the overall procedure
217  involves solving a linear number of linear programs and therefore runs in polynomial time. We can
218  repeat the procedure iteratively until no more actions are removed for any player. Thus, iterative

removal of dominated actions can be performed in polynomial time. Note that the procedure applies to all two-player games and does not assume that they are zero sum. The procedure also removes actions that are dominated by any mixed strategy (which may play a probability distribution over actions at the same information set $I$), not just actions that are dominated by a pure action.

Now suppose we want to determine if an action is dominated for player 1 in a game with $n$ players for $n > 2$. We can construct a new two-player game where player 2 now controls all actions that were previously controlled by any player other than player 1. Then we can run the above procedure in this new game. Thus, we can perform iterative removal of (strictly or weakly) dominated strategies in polynomial time in $n$-player extensive-form games of imperfect information.

**Theorem 1.** *There exists a polynomial-time algorithm that determines whether an action is strictly dominated in an $n$-player extensive-form game of imperfect information.*

**Theorem 2.** *There exists a polynomial-time algorithm that determines whether an action is weakly dominated in an $n$-player extensive-form game of imperfect information.*

**Theorem 3.** *Iterated removal of strictly and weakly dominated actions can be performed in polynomial time in $n$-player extensive-form games of imperfect information.*

## 5   Experiments

Now that we have defined dominated actions and showed that they can be computed efficiently, we would like to investigate whether they can be a useful concept in practice. Poker has been widely studied as a test domain for imperfect-information games. The most popular variant regularly played by humans is No-Limit Texas Hold'em (NLHE). Two-player NLHE works as follows. Initially two players each have a *stack* of chips. One player, called the *small blind*, initially puts $k$ worth of chips in the middle, while the other player, called the *big blind*, puts in $2k$. The chips in the middle are known as the *pot*, and will go to the winner of the hand. Next, there is an initial round of betting. The player whose turn it is to act can choose from three available options:

- *Fold:* Give up on the hand, surrendering the pot to the opponent.
- *Call:* Put in the minimum number of chips needed to match the number of chips put into the pot by the opponent. For example, if the opponent has put in \$1000 and we have put in \$400, a call would require putting in \$600 more. A call of zero chips is also known as a *check*.
- *Bet:* Put in additional chips beyond what is needed to call. A bet can be of any size from 1 chip up to the number of chips a player has left in their stack, provided it exceeds some minimum value and is a multiple of the smallest chip denomination. A bet of all of one's remaining chips is called an *all-in* bet (aka a *shove*).

The initial round of betting ends if a player has folded, if there has been a bet and a call, or if both players have checked. If the round ends without a player folding, then three public cards are revealed face-up on the table (called the *flop*) and a second round of betting takes place. Then one more public card is dealt (called the *turn*) and a third round of betting, followed by a fifth public card (called the *river*) and a final round of betting. If a player ever folds, the other player wins all the chips in the pot. If the final betting round is completed without a player folding (or if a player is all-in at an earlier round), then both players reveal their private cards, and the player with the best five-card hand (out of their two private cards and the five public cards) wins the pot (it is divided equally if there is a tie).

In some situations the blinds are very large relative to stack sizes. This can happen frequently at later stages in poker *tournaments*, where the blinds increase after a certain time duration. A common rule is that when stack sizes are below around 8 big blinds a *shove-or-fold* strategy should be employed where each player only goes all-in or folds [9]. Study of optimal shove-or-fold strategies has been considered for 2-player [10] and 3-player [3, 4] poker tournament endgames. The poker site Americas Cardroom[2] has specific "All-in or Fold" tables with up to 4 players where players are only allowed to play shove-or-fold strategies. The initial stack sizes at these tables are either 8 or 10 times the big blind; the highest stake available has blinds of \$100 and \$200 with initial stacks of \$2000.

In the two-player NLHE shove-or-fold game, each player has 169 strategically distinct hands with which they can choose to shove or fold (13 pocket pairs and $\frac{13 \cdot 12}{2} = 78$ combinations of each

---

[2]https://www.americascardroom.eu/

non-paired offsuit and suited hand). Let player 1 denote the small blind and player 2 denote the big blind. We assume that the blinds are small blind $k = 100$ and big blind $2k = 200$, and initial stacks are 1600 (8 times the big blind). We first remove all strictly dominated actions for player 1, followed by removing strictly dominated actions for player 2 (note that removing weakly dominated actions does not provide additional benefit in this game). It turns out that 85 actions for player 1 are removed and 99 actions for player 2 are removed. Thus, the initial game with 169 hands per player can be reduced to a game where player 1 must make a decision with 84 hands and player 2 must make a decision with 70 hands; the number of decision points has been reduced by over 50%. It turns out that performing an additional round of removing dominated actions does not remove any further actions.[3]

Next we consider the setting where the stacks are 5 times the big blind (i.e., 1000). In this game five rounds of iterated removal of dominated actions are needed. The first round removes 108 dominated actions for player 1 and 129 for player 2; the second round removes 20 for player 1 and 16 for player 2; the third round removes 8 for player 1 and 6 for player 2; the fourth round removes 7 for player 1 and 2 for player 2; the fifth round removes 1 for player 1 and 0 for player 2. In the final reduced game player 1 must make a decision with only 25 hands while player 2 must make a decision with only 16 hands. Table 5 shows the non-dominated actions for each player with parentheses indicating the iteration at which the alternative action was removed. 'S' indicated shove, 'F' indicates fold, and '?' indicates that neither action was dominated. If stacks are 4 times the big blind the game is solved completely after 4 rounds of removing dominated actions, and for stacks of 3 big blinds the game is solved completely after 2 rounds. These results demonstrate that iteratively removing dominated actions can significantly reduce the size of realistic games. While for simplicity we considered two-player zero-sum games, for which the full game can be solved directly by a linear program, the computational benefit for games with more than two players could be much more significant.

| | A | K | Q | J | T | 9 | 8 | 7 | 6 | 5 | 4 | 3 | 2 |
|---|---|---|---|---|---|---|---|---|---|---|---|---|---|
| A | S(1) | S(1) | S(1) | S(1) | S(1) | S(1) | S(1) | S(1) | S(1) | S(1) | S(1) | S(1) | S(1) |
| K | S(1) | S(1) | S(1) | S(1) | S(1) | S(1) | S(1) | S(1) | S(1) | S(1) | S(1) | S(1) | S(1) |
| Q | S(1) | S(1) | S(1) | S(1) | S(1) | S(1) | S(1) | S(1) | S(1) | S(1) | S(1) | S(1) | S(1) |
| J | S(1) | S(1) | S(1) | S(1) | S(1) | S(1) | S(1) | S(1) | S(1) | S(1) | S(1) | S(3) | ? |
| T | S(1) | S(1) | S(1) | S(1) | S(1) | S(1) | S(1) | S(1) | S(1) | S(1) | S(1) | S(3) | ? |
| 9 | S(1) | S(1) | S(1) | S(1) | S(1) | S(1) | S(1) | S(1) | S(1) | S(3) | ? | ? | ? |
| 8 | S(1) | S(1) | S(1) | S(1) | S(1) | S(1) | S(1) | S(1) | S(2) | S(4) | ? | ? | ? |
| 7 | S(1) | S(1) | S(1) | S(1) | S(1) | S(3) | S(5) | S(1) | S(2) | S(3) | ? | ? | F(3) |
| 6 | S(1) | S(1) | S(1) | S(1) | ? | ? | ? | ? | S(1) | S(3) | ? | ? | F(4) |
| 5 | S(1) | S(1) | S(1) | S(1) | ? | ? | ? | ? | ? | S(1) | S(4) | ? | ? |
| 4 | S(1) | S(1) | S(1) | S(2) | F(4) | F(2) | F(2) | F(3) | F(4) | ? | S(1) | ? | F(4) |
| 3 | S(1) | S(1) | S(1) | ? | F(2) | F(2) | F(2) | F(2) | F(2) | F(2) | F(2) | S(1) | F(3) |
| 2 | S(1) | S(1) | S(1) | F(4) | F(2) | F(2) | F(2) | F(2) | F(2) | F(2) | F(2) | F(2) | S(1) |

| | A | K | Q | J | T | 9 | 8 | 7 | 6 | 5 | 4 | 3 | 2 |
|---|---|---|---|---|---|---|---|---|---|---|---|---|---|
| A | S(1) | S(1) | S(1) | S(1) | S(1) | S(1) | S(1) | S(1) | S(1) | S(1) | S(1) | S(1) | S(1) |
| K | S(1) | S(1) | S(1) | S(1) | S(1) | S(1) | S(1) | S(1) | S(1) | S(1) | S(1) | S(1) | S(1) |
| Q | S(1) | S(1) | S(1) | S(1) | S(1) | S(1) | S(1) | S(1) | S(1) | S(1) | S(1) | S(1) | S(1) |
| J | S(1) | S(1) | S(1) | S(1) | S(1) | S(1) | S(1) | S(1) | S(1) | S(1) | S(3) | S(3) | ? |
| T | S(1) | S(1) | S(1) | S(1) | S(1) | S(1) | S(1) | S(1) | S(2) | ? | ? | ? | ? |
| 9 | S(1) | S(1) | S(1) | S(1) | S(1) | S(1) | S(1) | S(1) | S(3) | ? | F(4) | F(3) | F(2) |
| 8 | S(1) | S(1) | S(1) | S(1) | S(1) | S(3) | S(1) | S(1) | S(1) | ? | F(2) | F(2) | F(1) |
| 7 | S(1) | S(1) | S(1) | S(2) | ? | ? | ? | S(1) | S(1) | S(1) | F(2) | F(1) | F(1) |
| 6 | S(1) | S(1) | S(1) | ? | ? | ? | F(3) | F(2) | S(1) | S(1) | F(2) | F(1) | F(1) |
| 5 | S(1) | S(1) | S(1) | ? | F(4) | F(2) | F(2) | F(2) | F(1) | S(1) | S(1) | F(1) | F(1) |
| 4 | S(1) | S(1) | S(1) | ? | F(2) | F(2) | F(1) | F(1) | F(1) | F(1) | S(1) | F(1) | F(1) |
| 3 | S(1) | S(1) | S(1) | ? | F(2) | F(1) | F(1) | F(1) | F(1) | F(1) | F(1) | S(1) | F(1) |
| 2 | S(1) | S(1) | S(1) | F(2) | F(2) | F(1) | F(1) | F(1) | F(1) | F(1) | F(1) | F(1) | S(1) |

Figure 4: Dominated actions in 2-player NLHE allin-fold with 5 big blind stacks (player 1 left, player 2 right). Suited hands are in the upper right and unsuited hands are in the lower left.

# 6 Conclusion

Dominance is a fundamental concept in game theory. It is well-understood in strategic-form games, but its impact in imperfect-information games has so far been unexplored. We consider several plausible definitions of dominated actions which we demonstrate to be problematic; however, one of them which we denote as *strong domination* can be useful as an efficient preprocessing step. We present a new definition that addresses these limitations. We show that both strictly and weakly dominated actions can be identified in polynomial time, and that iterative removal can be performed in polynomial time in $n$-player games. Our algorithms identify actions that are dominated by any mixed strategy, not necessarily a pure action. We demonstrate empirically that removing dominated actions can play a significant role in reducing the size of realistic imperfect-information games. This can serve as an efficient preprocessing step before computation of a Nash equilibrium. In practice our algorithms could be sped up by several heuristics such as traversing the information sets in decreasing order of their depth. Recent work has shown that some games contain many "mistake" actions that are played with probability zero in all Nash equilibria but are not dominated [2]. Thus, there is potentially more future ground to explore on efficient game reduction by elimination of poor actions.

---

[3]We note that no actions are strongly dominated for either player in this game (i.e., Flawed Definition 1). For example, consider the action of player 1 folding with pocket aces (the best starting hand). The payoff of this action is -$100. If player 1 goes all-in, there are leaf nodes where player 1 will receive a payoff of -$1600.

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
