# OpenReview forum: "Dominated Actions in Imperfect-Information Games"
_NeurIPS.cc/2025/Conference — Submitted to NeurIPS 2025_

### Official Review · Reviewer_jQER · 2025-06-11

**Clarity:** 4
**Significance:** 2
**Originality:** 4
**Rating:** 3
**Confidence:** 3

**Summary:**

This paper presents a novel concept of "dominated actions" in extensive-form games. In normal-form games, dominated strategies are strategies that one can eliminate from consideration because playing another strategy is always better. In extensive-form games, there has been no widely accepted definition of dominated actions. The paper is an effort to fill this gap by defining dominated actions in extensive-form games and providing a method to compute them. The authors first propose several "flawed definitions" of dominated actions, explain why they are flawed, and then present their own definition. Empirically, the authors show that in the "All In or Fold" No-Limit Texas Hold'em poker variant, a significant number of actions are dominated, demonstrating the potential of this concept in practical applications.

**Questions:**

- Are the above weaknesses proper criticisms of the paper?

## Minor Comments

- In Lines 18 and 24, I believe the correct notation should be $u_i(\sigma_i',s_{-i}) > u_i(\sigma_i,s_{-i})$.

**Ethical Concerns:**

["NO or VERY MINOR ethics concerns only"]

**Final Justification:**

I have read the author response as well as the other reviews.

My central concern is that the paper's contribution seems slim. I appreciate the conceptual contribution of the work, and the author response has clarified the meaning of the paper's dedicated portion to previous flawed definitions and emphasized the algorithmic contributions. That being said, it does not resolve my doubt on the utility of this new idea. This concern is also present in other reviews, even in the most positive one by Reviewer Ztv9. It would be nice to see the potential of the proposed concept in developing new theoretical research or in solving large, more practical games.

In light of the above, my recommendation remains the same. If there is a 3.5 rating, I am willing to change my score to it.

**Limitations:**

I think the authors adequately mentioned the limitations.

**Paper Formatting Concerns:**

I did not find any paper formatting concerns.

**Quality:**

2

**Strengths And Weaknesses:**

# Strengths

- The idea of defining dominated actions in extensive-form games is novel and quite interesting.
- The paper is written in a way that is accessible to a broad audience, including those who may not be experts in game theory.
- The authors provide clear explanations with examples of why some definitions of dominated actions in extensive-form games are flawed.

# Weaknesses

- The contribution of the paper is novel, but slim. It seems to me that the proposal of the concept is the sole contribution. For example, more than half of this paper is dedicated to discussing flawed definitions of dominated actions and background information, which does not directly contribute to the main idea of the paper. As a result, the paper appears to be more of a position paper than a research paper.
- The empirical results are limited to a single game variant (the "All In or Fold" No-Limit Texas Hold'em poker variant), which may not generalize to other extensive-form games. Thus, the empirical evaluation is not very convincing to me, and I remain skeptical about the exact impact of having dominated actions in extensive-form games that people care about.

---

> ### Author Rebuttal · Authors · 2025-07-29
>
> "The idea of defining dominated actions in extensive-form games is novel and quite interesting."
>
> Thank you. Reviewer EP7V writes "The issue I have with the paper is that concepts and methods are all trivial and not particularly interesting or novel." Perhaps you two can discuss this because you seem to disagree.
>
> "The contribution of the paper is novel, but slim. It seems to me that the proposal of the concept is the sole contribution. For example, more than half of this paper is dedicated to discussing flawed definitions of dominated actions and background information, which does not directly contribute to the main idea of the paper. As a result, the paper appears to be more of a position paper than a research paper."
>
> That is an interesting take. I agree that coming up with the correct formulation of dominated actions in imperfect-information games is a key contribution of the paper. I think if I just stated the correct definition without providing context, examples, and demonstrating flaws with alternative definitions, the important nuances would be missed. I also demonstrate that whatever definition is being used by Gambit's method is flawed, and this is the only prior work on this topic I am aware of. I don't think that too much of the paper is spent on this.
>
> I disagree that the proposal of the concept is the sole contribution. I think the analysis in Section 4 that show that it can be computed in polynomial time for multiplayer games (for iterated dominance and for dominance by mixed strategies) are also major contributions. The experiments also demonstrate that the concept can be useful in realistic games.
>
> "The empirical results are limited to a single game variant (the "All In or Fold" No-Limit Texas Hold'em poker variant), which may not generalize to other extensive-form games. Thus, the empirical evaluation is not very convincing to me, and I remain skeptical about the exact impact of having dominated actions in extensive-form games that people care about."
>
> I am not sure which games you care about, but it is likely that some of them have very few dominated actions, and some have very many, similarly to dominated strategies in normal-form games. Regardless of the specific class of games considered, dominance is one of the most central concepts in game theory, and imperfect-information games have many important applications.

---

> > ### Comment · Reviewer_jQER · 2025-08-05
> >
> > I have read the author response as well as the other reviews.
> >
> > My central concern is that the paper's contribution seems slim. I appreciate the conceptual contribution of the work, and the author response has clarified the meaning of the paper's dedicated portion to previous flawed definitions and emphasized the algorithmic contributions. That being said, it does not resolve my doubt on the utility of this new idea. This concern is also present in other reviews, even in the most positive one by Reviewer Ztv9. It would be nice to see the potential of the proposed concept in developing new theoretical research or in solving large, more practical games.
> >
> > In light of the above, my recommendation remains the same. If there is a 3.5 rating, I am willing to change my score to it.

---

> ### Author Response · Authors · 2025-08-06
> **Response to comment**
>
> "It would be nice to see the potential of the proposed concept in developing new theoretical research or in solving large, more practical games."
>
> For normal-form games, many well-studied games have a large number of (iteratively) dominated strategies, e.g., Traveler's dilemma, Bertrand competition, centipede game, Guess 2/3 the average (aka beauty contest game). Similarly, for imperfect-information games, many games of all types will also have a large number of iteratively dominated actions. We presented a realistic poker variant as a conceptual example. I'm sure one could easily construct games from many important classes, e.g., a security game, for which many dominated or iteratively dominated actions exist. (In fact this could be done trivially by taking an existing game and adding in a large number of dominated actions to it). Dominated strategies are one of the most fundamental and central concepts in the study of normal-form games in game theory, and naturally dominated actions are similarly significant in extensive-form games.
>
> Although the specific poker we solved is relatively small, I think it clearly demonstrates the potential for the concepts to be useful in larger (and more practical) games. If we considered a similar game with larger deck size, or more players, we would see a similarly massive reduction in the number of decision points as a result of applying our procedure for removing dominated actions. Similarly if we are trying to solve a large 3 or 4-player security game that has many (iteratively) dominated actions, it may be totally intractable to solve the initial problem, but feasible to solve the reduced game after applying our procedure.

---

> > ### Comment · Reviewer_jQER · 2025-08-06
> >
> > > Although the specific poker we solved is relatively small, I think it clearly demonstrates the potential for the concepts to be useful in larger (and more practical) games.
> >
> > This potential is a bit unclear for me. I would be curious
> > - The specific poker variant you use is indeed too small to be practical. What if the game is larger, e.g., Leduc poker?
> > - Does this method help, when the game tree itself is too large to be traversed, e.g., no-limit Texas poker?

---

> ### Author Response · Authors · 2025-08-06
>
> "The specific poker variant you use is indeed too small to be practical. What if the game is larger, e.g., Leduc poker?"
>
> I agree that the game we experimented on is small, but I don't agree that it is "too small to be practical." The variant is played for significant amounts of real money daily on some poker sites (I referenced one in the paper). Ironically, while Leduc Holdem is larger than the variant we looked at, it is not a variant that people actually play. So one could argue that Leduc Holdem is "less practical" than All-In-Or-Fold Texas Holdem. Regardless, I have not run the game on Leduc Holdem so I could only guess how many actions would be dominated. I imagine the number would be moderate (folding most strong hands and calling with most weak hands would likely be dominated actions). In Kuhn poker, a smaller but similar variant, there are 12 total decision points in the full game, and only 5 after iterative removal of dominated actions. I imagine the percentage of dominated actions in Leduc would be similar.
>
> I am not sure whether the size of the game is directly tied to the number of dominated actions. There can be very large games with no dominated actions (e.g., if there is a unique Nash equilibrium that is fully mixed), and also large games for which 90% of actions are iteratively dominated.
>
> "Does this method help, when the game tree itself is too large to be traversed, e.g., no-limit Texas poker?"
>
> All of our definitions and algorithms are specifically extensive-form games of imperfect information that are represented in tree form. I think the concept of dominated actions could be extended to other game representations, but those representations would need to be defined explicitly. It is worth noting that many approaches for no-limit Texas Holdem work by first abstracting the game to a smaller version that can be represented as a game tree, though this tree is still too large to apply LP-based approaches and typically solved using variants of self-play counterfactual regret minimization with sampling. Regardless, this is a bit of an aside, and for the purposes of this paper we are focusing exclusively on the standard extensive-form imperfect-information tree representation. Perhaps future work could explore other game representations.

---

> > ### Comment · Reviewer_jQER · 2025-08-07
> >
> > Thank you for your detailed response. I think we generally share a similar understanding now, and I can imagine that some of your guesses might be true. For the paper in its current form, I think I will maintain my evaluation. I'd be willing to accept it if a future revision could indeed conduct the experiments on larger games like Leduc poker, and incorporate our discussion about the significance of this interesting idea.

---

### Official Review · Reviewer_yMbT · 2025-06-26

**Clarity:** 2
**Significance:** 3
**Originality:** 3
**Rating:** 3
**Confidence:** 4

**Summary:**

The paper considers different notions of dominance in extensive-form games, going from too restrictive, to more useful, all while retaining computational efficiency. Experiments validate that the more permissive definitions indeed result in significant elimination of dominated actions.

**Questions:**

1. (Taken from above) On lines 224-226 you write "We can construct a new two-player game where player 2 now controls all actions that
were previously controlled by any player other than player 1. Then we can run the above procedure in this new game." Are there concerns related to imperfect recall when setting up a metaplayer that controls any player other than player 1? In particular, the different players being controlled might now have asymmetric information, so a meta-player would struggle with imperfect recall.

2. (Taken from above) There doesn't seem to be a proof of the fact that the proposed definition of dominance indeed implies that the action can be removed safely from the tree. Is it immediately obvious that doing so is sound?

3. (Taken from above) In the experiments, you write that "In the two-player NLHE shove-or-fold game, each player has 169 strategically distinct hands." Even assuming taking suit isomorphism into account, the number of distinct hands would be much higher, because information about the multiplicity of suits, in relation to the public cards, must be taken into account. What am I missing?

4. The checklist contains problematic claims (see points 4a-4d above). Perhaps more importantly, echoing point 4a, the authors do not justify the lack of public/open source release (question 5 of the checklist). What is the reason for not providing code reproducing the experiments alongside the paper, especially when the paper's contribution is that of defining a notion of dominance that can be turned into an algorithm?

5. Currently, in two-player zero-sum games, the cost of removing dominance is similar to the cost of computing a Nash equilibrium. What do you imagine the computational benefits of dominance elimination will be?

6. I can see why the all-in or fold variant of poker would lend itself very well to the removal of dominated actions. In that sense, I think it was a very nice choice to highlight how your definition is better than the flawed definition. However, I am wondering how many actions are dominated in a more regular variant of poker (even something simple like fold-pot raise or limit hold'em). Do you have any sense for what the answer is?

7. Just to confirm: on lines 114-116, discussing the flawed definitions and setting the stage for your proposed definition, you say "The problem with these definitions are that they are too strong; it is still possible for player 1 to strictly prefer to take action bi to ai regardless of the strategy used by player 2 even if Flawed Definition 1 holds." Did you mean to write "even if Flawed Definition 1 does NOT hold"?

**Ethical Concerns:**

["NO or VERY MINOR ethics concerns only"]

**Final Justification:**

I thank the authors for their time and engagement in the discussion. I believe that by including a discussion on points 1 and 2 of the review, as promised by the authors, will make the paper stronger. I am slightly disappointed by the evasive answers of the authors around what I think was an expected and reasonable justification for why they are unwilling to release code, but overall I don't have any major reservations with the paper. I don't think that their lack of justification should be, by itself, a reason for rejection.

**Limitations:**

Yes.

**Quality:**

3

**Strengths And Weaknesses:**

I like the idea of the paper, and it was interesting to consider the different flawed definitions in the buildup. While I think the paper has an unfortunate typo around lines 114-116 (see below), I found the paper well organized.

My biggest reservation with the paper is along soundness and experimental setup. For soundness:
1. On lines 224-226 you write "We can construct a new two-player game where player 2 now controls all actions that
were previously controlled by any player other than player 1. Then we can run the above procedure in this new game." Are there concerns related to imperfect recall when setting up a metaplayer that controls any player other than player 1? In particular, the different players being controlled might now have asymmetric information, so a meta-player would struggle with imperfect recall.
2. There doesn't seem to be a proof of the fact that the proposed definition of dominance indeed implies that the action can be removed safely from the tree. Is it immediately obvious that doing so is sound?

For the experimental setup:

3. In the experiments, you write that "In the two-player NLHE shove-or-fold game, each player has 169 strategically distinct hands." Even assuming taking suit isomorphism into account, the number of distinct hands would be much higher, because information about the multiplicity of suits, in relation to the public cards, must be taken into account. What am I missing?

In addition, I am confused about the authors' checklist:

- 4a. The authors do not justify the lack of public/open source release (question 5 of the checklist). What is the reason for not providing code reproducing the experiments alongside the paper, especially when the paper's contribution is that of defining a notion of dominance that can be turned into an algorithm? The release of a reference implementation from the authors should be expected, in my view.

- 4b. The authors claim that they discuss the type of compute workers, memory, time of execution (question 8 of the checklist), but I could not find any such discussion referenced in the body.

- 4c. The authors claim to have "described safeguards that have been put in place for responsible release of data or models that have a high risk for misuse (e.g., pretrained language models, image generators, or scraped datasets)". However, no such risks were identified (and I don't think they exist).

- 4d. The authors do not release code, but claim to have provided documentation for their assets (question 13 of the checklist). Was this filled out by mistake?

Other comments:
- L18, Introduction: typo in the mathematical definition of domination
- L114-116. You say "The problem with these definitions are that they are too strong; it is still possible for player 1 to strictly prefer to take action bi to ai regardless of the strategy used by player 2 even if Flawed Definition 1 holds." Did you mean to write "even if Flawed Definition 1 does NOT hold"?
- L114: "The problem with these definitions are that they are too strong": It should be "The problem ... is that" (singular / plural)
- L116: "The is illustrated in the following game depicted" -> I would remove "following" for clarity
- Equation (8): missing period at the end of equation
- Reference [9]: the reference appeared malformed.

---

> ### Author Rebuttal · Authors · 2025-07-29
>
> "On lines 224-226 you write "We can construct a new two-player game where player 2 now controls all actions that were previously controlled by any player other than player 1. Then we can run the above procedure in this new game." Are there concerns related to imperfect recall when setting up a metaplayer that controls any player other than player 1? In particular, the different players being controlled might now have asymmetric information, so a meta-player would struggle with imperfect recall."
>
> I am assuming that the game has perfect recall. I describe this assumption in Section 2, but perhaps can make it more clear that this is the setting we are considering. Many things can break down in imperfect-recall games (e.g., Nash equilibrium is no longer guaranteed to exist). It would be an interesting future avenue to consider dominated actions in that setting, but for our paper we assume there is perfect recall.
>
> "There doesn't seem to be a proof of the fact that the proposed definition of dominance indeed implies that the action can be removed safely from the tree. Is it immediately obvious that doing so is sound?"
>
> Yes the argument is similar as the one for normal-form games. If an action A is (strictly) dominated according to our definition, then there exists another mixed strategy that plays A with probability 0 at that information that always does strictly better. So no equilibrium can follow A assuming the information set is reached, and we are always guaranteed that a Nash equilibrium exists in the reduced game. I wouldn't necessarily say that it is immediately obvious, but it seems essentially the same argument as for the normal-form case.
>
> "In the experiments, you write that "In the two-player NLHE shove-or-fold game, each player has 169 strategically distinct hands." Even assuming taking suit isomorphism into account, the number of distinct hands would be much higher, because information about the multiplicity of suits, in relation to the public cards, must be taken into account. What am I missing?"
>
> In the shove-or-fold game, both players decide to shove or fold prior to any of the public cards being dealt. If player 1 shoves and player 2 calls, then all 5 public cards are dealt out. But the strategies themselves are only conditioned on each player's 2 private cards and not the public cards.
>
> 4) I may decide to release my code publicly if the paper is accepted.
>
> 5) "Currently, in two-player zero-sum games, the cost of removing dominance is similar to the cost of computing a Nash equilibrium. What do you imagine the computational benefits of dominance elimination will be?"
>
> There may not be computational benefits of dominance elimination in two-player zero-sum games. Perhaps the algorithm or implementation can be optimized in such a way that the linear programs are significantly smaller than the full one. But to give a better answer I have two responses. First, dominated actions are also of interest on their own beyond Nash equilibrium. Removing them as a preprocessing step for Nash equilibrium is only one motivation. Second, I think the main computational benefit is for non-zero-sum and multiplayer games, since computing a Nash equilibrium is PPAD and computationally intractable for even relatively small games, while our procedures run in polynomial time.
>
> 6) "I can see why the all-in or fold variant of poker would lend itself very well to the removal of dominated actions. In that sense, I think it was a very nice choice to highlight how your definition is better than the flawed definition. However, I am wondering how many actions are dominated in a more regular variant of poker (even something simple like fold-pot raise or limit hold'em). Do you have any sense for what the answer is?"
>
> Larger versions of poker are more complicated and can exhibit a wide range of behaviors. I don't want to get too much into it because of anonymity. In some poker games only folding the best possible hand or calling with the worst possible hand are dominated. But in games with small deck sizes like 2 and 3-player Kuhn poker a lot of actions are dominated. As for dominance in normal-form games, I think real games will exhibit a wide range of the number of dominated actions. Our experiments demonstrate that the concept is relevant in a natural game (that is actually played).
>
> "Just to confirm: on lines 114-116, discussing the flawed definitions and setting the stage for your proposed definition, you say "The problem with these definitions are that they are too strong; it is still possible for player 1 to strictly prefer to take action bi to ai regardless of the strategy used by player 2 even if Flawed Definition 1 holds." Did you mean to write "even if Flawed Definition 1 does NOT hold"?"
>
> I mean that if we are using Flawed Defintiion 1 as the actual definition of dominated actions, then it would be problematic because it would be possible that player 1 could strictly prefer to take action b_i to a_i regardless of player 2's strategy, but the definition would not say that a_i is dominated. This flaw is demonstrated in the game in Figure 1. I can reword this sentence if it is confusing.

---

> > ### Author Response · Authors · 2025-08-01
> > **Edit to rebuttal for lines 114-116**
> >
> > Edit to my rebuttal: you are correct for lines 114-116 it should be even if the definition does not hold. Thank you.
> >
> > Also thank you for the other typos you found.
> >
> > I am not sure about the missing period in Equation (8)?
> >
> > I believe Reference 9 was correct at the time at least. That link seems to no longer work at the moment (requires a user login now). The name of the author was listed as username mersenneary with lowercase first letter.

---

> > ### Comment · Reviewer_yMbT · 2025-08-07
> >
> > Thank you for the clarifications. I have a couple of follow-up points.
> >
> > 1. About lines 224-226. Perhaps I didn't understand the answer, but I think that my concern has not been addressed. My concern is the following: the players that are merged into a single meta-player might now make the meta-player have imperfect recall, even if the original game (before merging players) was perfect recall. This is because the different players that are being merged might have had perfect recall but access to different imperfect information. Perhaps an example on 3-player poker can elucidate my point. Player 2 and player 3 each have perfect recall individually. Each has access to their own private hand before making decisions. A meta-player that controls the two players now needs to be careful about whether the meta-player, taking what earlier was player 3's action, can condition on player 2's private hand. Indeed, since the meta-player now controls both player 2 and player 3, the meta-player now has observed player 2's hand. Another way of saying this is: unless the meta-player is made imperfect-recall, the set of strategies available to the meta-player will now leak information  that was precluded before merging the players, resulting in correlation between the actions of the merged players. My question is: should this subtlety be a concern?
> >
> > 2. Thank you. I think it would be important to provide a formal proof (in more detail than the rebuttal) of this fact in the paper.
> >
> > 3. Thanks for the clarification. I am not familiar with shove-or-fold. Could you please clarify how many information sets the game tree has?
> >
> > 4. My concerns regarding the way the checklist has been filled out have not been addressed. As per guidelines, the checklist is an important part of the submission (and in fact, papers have been desk rejected in the past based on a missing checklist for example). In my review I pointed out that several questions in the checklist were missing or contradictory. See points 4b-4d in my review. Perhaps most concerning was question 5 in the checklist (point 4a). The checklist asks for a justification for why the authors are refusing to release code. In an attempt to solicit the missing answer to the question, question 4 in my review was: "What is the reason for not providing code reproducing the experiments alongside the paper, especially when the paper's contribution is that of defining a notion of dominance that can be turned into an algorithm?". The author's response, "I may decide to release my code publicly if the paper is accepted", is not an answer to my question, and fails again to provide a justification to the checklist.
> >
> > 5. Thanks for the answer. I agree with your argument.
> >
> > 6. Thanks. If you have any data points, could you please elaborate on what you mean when you say "wide range of behavior"? I think including a discussion on this point in the paper would be an interesting addition.
> >
> > 7. Thanks for confirming regarding lines 114-116. And for confirming Reference 9 is displayed as intended and not malformed. You are right that Equation (8) should not have a period, to keep it in line with the style of the other equations.

---

> > > ### Author Response · Authors · 2025-08-07
> > >
> > > 1) The meta-player is aware of all the private information to both players 2 and 3 in your example, so there would still be perfect recall. I agree that this new scenario is fundamentally different from the cases when players 2 and 3 had different private information; however, this does not matter according to the definition of dominated actions (definition 1-2). The requirement is that the action leads to higher expected utility regardless of the strategies played by the opposing players, which may or may not be "correlated."
> > >
> > > 2) I can add this.
> > >
> > > 3) Each player has 169 different decision nodes, so there are 338 information sets (ignoring chance moves where no player takes an action). There are 169 strategically distinct preflop hands. With a given hand, player 1 must decide to shove or fold (without knowing player 2's hand). If player 1 folds, then the game ends. If player 1 shoves, then player 2 must decide to call or fold. If player 2 calls, then the board is run out (without any further action by player 1 or 2).
> > >
> > > 4) I am looking over the instructions for the checklist:
> > >
> > > "While we encourage release of code and data, we understand that this might not be possible, so no is an acceptable answer. Papers cannot be rejected simply for not including code, unless this is central to the contribution (e.g., for a new open-source benchmark)."
> > >
> > > I do not believe that I am required to include code, or to justify why I may or may not want to include code. Perhaps the area chair can correct me if I am misinterpreting the instructions.
> > >
> > > 6) Do you mean for poker games or in general? In some of the other responses I gave some specific (normal-form) games that have many dominated strategies, and some that have very few: e.g., Traveler's dilemma, Bertrand competition, centipede game, Guess 2/3 the average (aka beauty contest game) have many (iteratively) dominated strategies, while games like rock-paper-scissors have none. In Kuhn poker there are 12 decision nodes, and only 5 after iterative removal of dominated actions. I think Leduc holdem would be similar. In full Texas Holdem I suspect that it would be quite hard for an action to be dominated, so the number is probably small. E.g., if you are facing a large bet on the river with the 2nd worst-possible hand, there is still a possibility that the opponent is only betting that amount with the worst hand, so calling would not be dominated. This relates to the final sentences in the conclusion about future work. I think there is more interesting work to be done beyond dominance for identifying poor actions.

---

> > > > ### Comment · Reviewer_yMbT · 2025-08-07
> > > >
> > > > Thank you for your response. I don't have further questions.
> > > >
> > > > I think we leave point 4 unresolved. My goal was to understand why the authors are unwilling (or unable) to release code, particularly for a paper proposing a general-purpose algorithm that could help other papers. I believe a reason is expected when answering "No" to question 5; the AC may be able to clarify this. Mandatory or not, I would have appreciated some explanation from the authors. But I think I've made my point and don't want to push further.
> > > >
> > > > Thank you again for your time and for responding to my points!

---

### Official Review · Reviewer_Ztv9 · 2025-07-01

**Clarity:** 4
**Significance:** 3
**Originality:** 4
**Rating:** 5
**Confidence:** 3

**Summary:**

This paper investigates the dominance action in imperfect-information extensive-form games. While dominated strategies are well-understood in strategic-form games, their impact in extensive-form games remains largely unexplored. To address this, the authors first consider several plausible definitions of dominated actions and identify their limitations. They then propose a new formal definition of dominated actions suitable for extensive-form games, addressing these limitations. Then they develop polynomial-time algorithms to identify strictly and weakly dominated actions and show that iterative removal of such actions can also be performed in polynomial time for n-player games. Finally, empirical results on a No-Limit Texas Hold’em poker variant illustrate that iteratively removing dominated actions can effectively reduce the size of the game.

**Questions:**

1.	In the experimental section, to better evaluate the scalability of the proposed identification algorithm, could the authors provide the number of variables and constraints of the linear programming problem used in each identification step, as well as the total number of information sets and actions in the tested game, and the time required for each step?

2.	Since the proposed method is theoretically applicable to n-player imperfect-information extensive-form games, could the authors provide experimental results or at least a discussion for a simple multi-player setting to support the claimed generality? For example, even a small-scale three-player game would help illustrate the practical feasibility of extending the approach beyond the two-player case.

**Ethical Concerns:**

["NO or VERY MINOR ethics concerns only"]

**Final Justification:**

For my questions, the authors have provided clear responses. For the scalability issue, I think the response from the authors can be accepted, as it does not intend to solve a large-scale game directly. Removing dominated actions would help reduce the game size. Therefore, I maintain my previous rating.

**Limitations:**

Yes

**Quality:**

3

**Strengths And Weaknesses:**

This paper focuses on a very interesting topic: dominated actions in imperfect-information games. To the best of my knowledge, prior work has largely overlooked this specific problem, making the contribution both novel and relevant to the game theory community. The introduction of the definition of dominated action in Section 3 is clearly presented, and the discussion of several flawed definitions is helpful for motivating and understanding the final formal definition proposed by the authors.

The algorithm for identifying dominated actions involves solving a linear number of linear programs. Although this is polynomial in theory, its practical scalability to larger and more realistic imperfect-information extensive-form games remains uncertain. Furthermore, the experimental section is limited to relatively small-scale games, making it unclear whether the approach would remain computationally feasible in settings with thousands of information sets and actions.

Mirror Comment: In Lines 115–116, it seems the intended phrasing should be “even if Flawed Definition 1 does not hold” rather than “even if Flawed Definition 1 holds”, to match the logical argument of the surrounding discussion.

---

> ### Author Rebuttal · Authors · 2025-07-29
>
> "Although this is polynomial in theory, its practical scalability to larger and more realistic imperfect-information extensive-form games remains uncertain. Furthermore, the experimental section is limited to relatively small-scale games, making it unclear whether the approach would remain computationally feasible in settings with thousands of information sets and actions."
>
> I don't think that the scalability would be a problem, since we can solve quite large two-player zero-sum games using similar linear programs from which our approaches are based. I think the bigger issue is that some games may have very few dominated actions, in which case our approaches would not be helpful.
>
> "In the experimental section, to better evaluate the scalability of the proposed identification algorithm, could the authors provide the number of variables and constraints of the linear programming problem used in each identification step, as well as the total number of information sets and actions in the tested game, and the time required for each step?"
>
> The purpose of our experiments was more to show the potential existence of a large number of (iteratively) dominated actions in a realistic imperfect-information game, which validates the usefulness of the concept. I admit that the example games are very small and not indicative of scalability. As described, there are 169 different information sets/decision points for each player, and the algorithm runs in less than a second. To get a sense of scalability, note that we are solving linear programs that are relatively similar to the sequence-form LP formulation which we discuss, which I believe has been successfully applied to games with 10^9 nodes.
>
> "Since the proposed method is theoretically applicable to n-player imperfect-information extensive-form games, could the authors provide experimental results or at least a discussion for a simple multi-player setting to support the claimed generality? For example, even a small-scale three-player game would help illustrate the practical feasibility of extending the approach beyond the two-player case."
>
> I agree that the main benefit of the approach would come for games that are two-player non-zero-sum and multiplayer, since two-player zero-sum games can already be solved in polynomial time by an LP (see response above to Reviewer EP7V). But the purpose of the experiments is not to demonstrate the computational efficiency, but to show the relevance of the concept of dominated actions in realistic games. It seems clear that there likely also exist many multiplayer games which contain a large number of (iteratively) dominated actions as well (though of course there also exist games with very few dominated actions). For multiplayer games we are still solving a linear number of linear programs, each of which resembles the sequence-form LP. This should be efficient and scalable in practice similarly to its success in two-player zero-sum games. I don't think I'm allowed to promise new work in the rebuttal, but if the paper is accepted I will make an effort to add in results on a multiplayer game in an additional page.

---

> > ### Comment · Reviewer_Ztv9 · 2025-08-05
> >
> > Thanks for the authors' responses. The responses somehow clarified my concerns.
> >
> > However, I still find the scalability issue somewhat unresolved. Specifically, the response states that “I don't think that the scalability would be a problem and the bigger issue is that some games may have very few dominated actions, in which case our approaches would not be helpful.” This appears to acknowledge that in large-scale games with few dominated actions, the proposed method may yield limited benefits. Consequently, the scalability would be a problem, as the method may not meaningfully reduce the complexity in such cases. It would be helpful if the authors could further clarify the scope of applicability of their approach.

---

> > > ### Author Response · Authors · 2025-08-06
> > > **Re: scalability**
> > >
> > > Hi, the issue that I mentioned that you reiterate is not a scalability issue. It is the fact that only some games will have some/many dominated actions, while others will not. This is also true for dominated strategies in normal-form games. For example, suppose that a game contains one Nash equilibrium that is totally mixed (puts positive probability on all pure strategies in a normal-form game). For example, rock-paper-scissors or its generalizations. Then no strategies are dominated. On the other hand, in many classic games a large number of strategies are removed by iterative dominance, e.g., Traveler's dilemma, Bertrand competition, centipede game, Guess 2/3 the average (aka beauty contest game). Nonetheless, strategic dominance is considered one of the most fundamental concepts in game theory, despite the fact that certain games do not possess any dominated strategies. This is true regardless of the size of the game.
> > >
> > > The same phenomenon would occur for dominated actions in imperfect-information games. Some games naturally would not have any (iteratively) dominated actions, and some games would have many dominated actions. The concept and algorithm have potential to be tremendously useful in many games, while they may not be helpful in certain games. In terms of scalability, removal of dominated actions could make Nash equilibrium tractable in games for which it is not currently. In the example poker game we presented (lines 278-291), the number of decision points in the game is reduced by 88% (from 169*2 = 338 to 25+16 = 41). This is particularly useful in non-zero-sum and multiplayer games. Suppose we had a 3-player game (for which Nash equilibrium computation is PPAD-hard) that is too large to solve for a Nash equilibrium, but if we apply our algorithms we reduce the size of the game by 88% and guarantee that the reduced game contains a Nash equilibrium of the original game. This would be extremely beneficial, because perhaps the smaller game is well-within the computational scale of equilibrium-finding algorithms while the original game is not.
> > >
> > > So to summarize, yes it is true that the concept may not be helpful in all games. But in many games it may make such a large difference that previously intractable games can now be solvable. This would be particularly useful for large games which are currently intractable to solve by existing current means. So I don't think there is a "scalability issue" with the approach; on the contrary, our concept and algorithms can be particularly useful for large games if they contain iteratively dominated actions.

---

> > > > ### Comment · Reviewer_Ztv9 · 2025-08-07
> > > >
> > > > Thanks for your clarification. Currently, it is clear to me about the scalability issue.
> > > >
> > > > I think it is more akin to an abstraction method applied to large-scale games, which is used to reduce the complexity or scale of the game. Perhaps this metaphor is not very appropriate.
> > > >
> > > > My current evaluation is already high, and I will maintain this score.

---

> ### Author Response · Authors · 2025-08-01
> **Edit response to lines 115-116**
>
> Edit to my rebuttal: you are correct for lines 115-116 it should be even if the definition does not hold. Thank you.

---

### Official Review · Reviewer_EP7V · 2025-07-03

**Clarity:** 4
**Significance:** 1
**Originality:** 1
**Rating:** 2
**Confidence:** 5

**Summary:**

The goal of the paper is to introduce a new notion/definition of dominated actions in extensive form games and provide an algorithm for finding such actions and consequently for iterated elimination of dominated actions. Authors discuss potential candidates for the definition of dominated actions and show their weaknesses. Authors propose a new definition that does not have the weaknesses of the previous candidates. Then they derive a general procedure based on linear programming for finding dominated actions in EFGs (also for more than two players, and general sum). Authors evaluate their algorithm for iterative removal of dominated actions in "all in or fold" no-limit texas holdem poker and show that their algorithm successfully removes dominated actions according to their definition.

**Questions:**

As you assume you would construct and run LP in each infoset, wouldn't the construction and the final LP be as big/hard as solving the original game?

**Ethical Concerns:**

["NO or VERY MINOR ethics concerns only"]

**Final Justification:**

I have thought deeply about the result. While I agree with some of the points raised by the authors, my main concern remains that I feel this result is trivial and I still think that's true.

**Limitations:**

Yes

**Quality:**

3

**Strengths And Weaknesses:**

I like that the authors explained well the motivation behind the definitions, including the “flawed” ones. The “flawed” definitions really help to get the reasoning process across well, and make a good motivation for the final Definitions (1) and (2). The included examples/figures are also useful and help to explain the concepts.

The issue I have with the paper is that concepts and methods are all trivial and not particularly interesting or novel.

The final definition basically compares expected value of “taking an action a at infoset I” and “not taking an action a at infoset I”. Computing these expectations via a sequence form LP representation is trivial and mechanical - and that’s exactly what the equations in the later section do. All that one then needs for the action to be dominated is if one expectation is greater than the other, regardless of the agent’s policies (as long as they reach the infoset). This is again trivial and mechanical operation in the sequence form LP representation (as a small note, I don’t think there’s a need to go through Lagrangian and dual to get to a point where we have both player doing max - one can just argue that for the inequality to hold, it needs to hold for the extreme values and these are simply achieved when both players do max)

Furthermore, the paper doesn’t include any discussion on prior work on this topic. My impression is that this work would classify more as an “exercise for a reader” in a game theory textbook rather than a submission to this venue.

---

> ### Author Rebuttal · Authors · 2025-07-29
>
> "The issue I have with the paper is that concepts and methods are all trivial and not particularly interesting or novel."
>
> I'm having some trouble understanding this comment. Domination is one of the main solution concepts in game theory and imperfect-information games have received a significant amount of study; if the concepts were obvious and/or trivial, then surely someone would have discovered them already. The only prior work I am aware of is Gambit's method that uses a flawed version of dominated actions (as I described in the paper). I have personally spent years thinking about the concept of dominated actions in imperfect-information games to some extent and have not obtained these results until now. It can be very easy to say that things are trivial or obvious in hindsight.
>
> Reviewer jQER writes:
> "The idea of defining dominated actions in extensive-form games is novel and quite interesting."
> Perhaps you two can discuss this because you seem to disagree.
>
> I don't understand what you mean when you say that the concepts are not novel. All of the concepts and algorithms are new. Do you mean that using linear program formulations, Lagrangians, and duality are not novel methods? I don't think that authors are required to invent a new "method."
>
> "Furthermore, the paper doesn’t include any discussion on prior work on this topic."
>
> This comment (in addition to your 5 confidence rating) seems to suggest that there is prior work on this topic that you are aware of that we failed to cite. It seems strange to make this comment without providing any references to such work. I cited the only prior work I am aware of on dominated actions in imperfect-information games -- the Gambit software package. See lines 170-179 on page 5. Gambit does not have any paper or documentation on the definition or approach that they used, but I demonstrated that the approach does not correctly remove all dominated actions. I am not aware of any other work on this topic. I also cited several prior works on dominated strategies in normal-form games.
>
> "As you assume you would construct and run LP in each infoset, wouldn't the construction and the final LP be as big/hard as solving the original game?"
>
> When you say "solving," I assume you are referring to computing one Nash equilibrium. While removing domination can be useful as a preprocessing step for Nash equilibrium computation, domination is a separate fundamental solution concept that may be of interest on its own. I agree that as presented the approach may not be practically useful for two-player zero-sum games, for which a Nash equilibrium can be found in polynomial time e.g., by an LP. But for two-player non-zero-sum games and multiplayer games it is PPAD-hard to compute one Nash equilibrium, and doing so in even small games can be computationally challenging. For these game classes, it could be very useful to apply a polynomial-time procedure that potentially provides a significant reduction in the game size before trying to compute an equilibrium. Also, even for the two-player zero-sum setting perhaps the algorithm could be improved or implemented in a way such that it involves solving LPs that are significantly smaller than the full one.

---

> > ### Author Response · Authors · 2025-08-06
> >
> > I would appreciate if the reviewer could provide references for the prior work on the topic that they are referring to.

---

### Note · Authors · 2025-08-12

Thank you to the reviewers.

Reviewer EP7V never replied to my rebuttal or comment. They claim "the paper doesn’t include any discussion on prior work on this topic," but failed to provide any relevant references that were omitted in response to my inquiry.

For other reviewers I believe most comments and questions were addressed in the subsequent comments.

---

### Decision · Program_Chairs · 2025-09-17

**Decision:**

Reject

**Comment:**

**Summary**

The main contribution of the paper is an algorithm for iteratively removing dominated actions in imperfect-information extensive-form games. The paper also provides an experimental evaluation of the proposed algorithm on some instances of No-Limit Texas Hold’em Poker. The main goal of the experimental evaluation is to show that the algorithm can be used as a pre-processing step to speed-up Nash equilibrium computation in two-player zero-sum imperfect-information extensive-form games.

**Strengths**

The problem addressed by the paper is interesting, and the solutions are clearly presented.

**Weaknesses**

The major weakness of the paper is the scalability of the proposed algorithms. All the definitions of dominated actions for imperfect-information games introduced in the paper should be better motivated.

**Decision**

I recommend **rejection** for the paper. All the Reviewers agreed on the scalability issues of the proposed algorithms, as well as their applicability in general. Some of the Reviewers recognized the merits of the dominance definitions and algorithms introduced in the paper. However, they all agreed that these are *not* sufficient to meet the acceptance bar at NeurIPS. I believe that the Authors should put additional effort in better assessing the scalability of the proposed algorithms.